# Time to diabetic neuropathy and its predictors among adult type 2 diabetes mellitus patients in Amhara regional state Comprehensive Specialized Hospitals, Northwest Ethiopia, 2022: A retrospective follow up study

**Sharie Tantigegn**[1], **Atsede Alle Ewunetie**[2], **Moges Agazhe**[2], **Abiot Aschale**[2], **Muluye Gebrie**[2], **Gedefaw Diress**[2], **Bekalu Endalew Alamneh**[2]*

1 Dega Damot District Health Office, West Gojjam, Feresbet, Ethiopia, 2 Department of Public Health, College of Health Sciences, Debre Markos University, Debre Markos, Ethiopia

* bekiehsm@gmail.com

## Abstract

### Background

Diabetic neuropathy is the primary cause of foot ulcers and amputations in both industrialized and poor countries. In spite of this, most epidemiological research on diabetic neuropathy in Ethiopia have only made an effort to estimate prevalence, and the information underlying the condition's beginning is not well-established. Therefore, determining the time to diabetic neuropathy and its variables among adult patients with type 2 diabetes mellitus at the Compressive Specialized Hospitals of the Amhara region was the aim of this study.

### Methods

An institutional-based retrospective follow-up study was undertaken among 669 newly recruited adult patients with type 2 diabetes mellitus who were diagnosed between the first of March 2007 and the last day of February 2012. Patients with diabetic neuropathy at the time of the diagnosis for type 2 diabetes mellitus (T2DM), patients without a medical chart, patients with an unknown date of DM diagnosis, and patients with an unknown date of diabetic neuropathy diagnosis were excluded from the study. All newly diagnosed type 2 diabetes mellitus (T2DM) patients aged 18 years and older who were enrolled from 1st March 2007 to 28th February 2012 in selected hospitals were included in this study. Cox proportional hazard model was fitted to determine predictors of time to diabetic neuropathy, and the Kaplan Meier survival curve was used to assess the cumulative survival time. Variables with a p-value < 0.05 were considered to be statistically significance at 95% confidence interval.

### Results

The restricted mean survival time of this study was 179.45 (95% CI: 173.77–185.14) months. The overall incidence rate of diabetic neuropathy was 2.14 cases per 100 persons-

**Data Availability Statement:** All relevant data are within the paper.

**Funding:** The author(s) received no specific funding for this work.

**Competing interests:** The authors have declared that no competing interests exist.

**Abbreviations:** AHR, Adjusted Hazard Ratio; BMI, Body Mass Index; CHR, Crude Hazard Ratio; DMCSH, Debre Markos Compressive Specialized Hospital; DTCSH, Debre Tabor Compressive Specialized Hospital; FHCSH, Felege Hiwot Compressive Specialized Hospital; HDL, High Density Lipoprotein; LDL, Low Density Lipoprotein; OPD, Out Patient Department; RMST, Restricted Mean Survival Time; T2DM, Type 2 Diabetes Mellitus; TC, Total Cholesterol.

years. Being aged > 60 years [AHR = 2.93(95% CI: 1.29–6.66)], having diabetic retinopathy [AHR = 2.76(95% CI: 1.84–4.16)], having anemia [AHR = 3.62 (95% CI: 2.46–5.33)], having hypertension [AHR = 3.22(95% CI: 2.10–4.93)], and baseline fasting blood sugar > 200 mg/dl [AHR = 2.56(95% CI: 1.68–3.92)] were the predictors of diabetic neuropathy.

## Conclusion

The risk of occurrence of diabetic neuropathy among type two diabetes mellitus patients was high in the early period. Age > 60 years, diabetic retinopathy, anemia, baseline fasting blood sugar level > 200 mg/dl, and hypertension were the main predictors of incidence of diabetic neuropathy. Therefore, early detection and appropriate interventions are important for patients with old age, diabetic retinopathy, anemia, hypertension, and FBS > 200mg/dl.

## Background

Diabetes mellitus is a group of metabolic disorders that affect the metabolism of fat, carbohydrates, and proteins and are defined by hyperglycemia brought on by impaired insulin production, insulin action, or both [1–3]. It is in charge of both sudden and recurring difficulties. Hypoglycemia, diabetic ketoacidosis, and hyperosmolar hyperglycemia syndrome are all part of the acute complication. Both macro-vascular complications, which affect big and medium-sized blood vessels, and micro-vascular complications, which impact small blood vessels, are classified as chronic complications [4, 5]. Atherosclerosis, cerebrovascular disease, stroke, hypertension, and coronary heart disease are among the macro-vascular complications of diabetes mellitus [6].

After all other causes have been ruled out, diabetic neuropathy (DN) is defined as any sort of nerve injury linked to diabetes [7]. Diabetic neuropathies occur in a variety of forms, including proximal, autonomic, focal, and peripheral neuropathy [8]. One of the most significant micro vascular consequences of DM and the most frequent reason for foot ulceration and amputation is diabetic neuropathy [9].

Diabetic Neuropathy (DN) is a global health care problem for both developed and developing countries [10]. It is estimated that every 30 seconds somewhere in the world, a lower limb or part of the lower limb is lost due to diabetes [11]. Diabetic Neuropathy accounts for 80% of foot ulceration [12] and 50–60% of non-traumatic limb amputations performed due to diabetic neuropathy and the problem is growing [13]. Globally, the pooled prevalence of diabetic neuropathy among patients with diabetes ranges from 22–46.5% [7, 14–16]. In Africa 22–66% of diabetic patients developed diabetic neuropathy [10, 17, 18] and in Ethiopia it ranges from 52.2–53.6% [19, 20]. The prevalence of diabetic neuropathy is high in developing countries due to late diagnosis, scarcity of screening and diagnosing resources, poor control of blood glucose, rise in the cost of health expenditures, poor medical facilities, and lack of adequate service for diabetic care [21, 22]. In addition to this patients with DN suffer from severe pain [23], frequent disturbance of sleep, decrease work productivity, increase the cost of treatment due to prolonged hospitalization [24], inability to work due to physical limitation and frequent hospitalization, patient quality of life seriously affected due to impact on social and psychological well-being of the individual [25].

The most common risk factors for the development of diabetic neuropathy are the duration of diabetes mellitus, age, body mass index, hypertension, diabetic retinopathy, and fasting blood sugar level [26–28].

Early recognition and detection of diabetic neuropathy, modification of risk factors, education of the patient, adequate glycemic control, keeping blood pressure under control, maintaining a healthy weight, and regular physical activities are strategies settled to reduce the complication and morbidity due to diabetic neuropathy [29, 30]. To date, most epidemiological research on diabetic neuropathy in Africa including Ethiopia has been limited to prevalence estimation from cross-sectional studies. Estimating the time to diabetic neuropathy and early detection of the risk factors is important for the prevention of DN. Therefore, this study aimed to determine time to diabetic neuropathy and its predictors among adult type 2 diabetes mellitus patients in Amhara region Compressive Specialized Hospitals, Northwest Ethiopia. The study's hypotheses were as follows:

1. Patients with type 2 diabetes mellitus have a different onset period for diabetic neuropathy.

2. Patients with type 2 diabetes mellitus are at risk for developing diabetes neuropathy due to certain risk factors.

## Methods

### Study design and settings

An institutional-based retrospective follow up study was conducted from 1st March 2007 to 28th February 2022 at Debre Markos Specialized hospital, Debre Tabor Specialized hospital and Felege-Hiwot Specialized Hospital. These hospitals are the referral site for chronic diseases in Amhara regional state.

The study participants were enrolled from 1 March 2007 to 28 February 2022, and the follow-up time was from their enrollment until the development of the event. Study participants who were lost to follow-up, died, transferred out before developing the event, or were event-free at the end of the study were censored. At the time of the initial T2DM diagnosis, the baseline data measurement was initiated. All newly diagnosed type 2 diabetes mellitus (T2DM) patients aged 18 years and above who were enrolled from 1st March 2007 to 28th February 2012 in selected hospitals were included in this study while Patients who had diabetic neuropathy at the time of the diagnosis for T2DM, patients who had no medical chart, patients with an unknown date of DM diagnosis, and patients with unknown date of diabetic neuropathy diagnosis were excluded from the study.

### Sample size and sampling procedures

The sample size was determined using the proportional allocation freedman principle formula $n = \frac{number\ of\ events}{Probability\ of\ event}$, Number of events $= \frac{(Z\alpha/2+Z\beta)^2}{pq(logHR)^2}$, Probability of the event = 1-(ps1 (t) +qs2 (t)) [31] by assuming 95% CI, 5% margin of error, HR (= 0.43) [27], design effect 1.5 and 80% power. The final sample size was 669. Simple random sampling method was used to select study participants. Initially, three specialized hospitals (30 percent of specialized hospitals found in the Amhara region) were selected using lottery method. Then, proportional allocation of sample was done among selected hospitals based their patient flow by under taking preliminary data ((371 (NDTCSH), 589 (NFHCSH), and 424 (NDMCSH)). Finally, study participants (nDTCSH = 179, nFHCSH = 285 and nDMCSH = 205) were selected by computer generated random sampling technique using their medical registration number as a sampling frame.

## Variables

Time to diabetic neuropathy were outcome variable of this study. Socio-demographic variables (sex, age, and place of residence), clinical variables (family history of complications of Diabetes mellitus, type of DM treatment ((oral(Glibenclamide)or injectable)), and family history of Diabetes mellitus), Biochemical variables (fasting blood glucose level at baseline, proteinuria, micro-albuminuria, triglyceride level, low-density lipoprotein level, high-density lipoprotein level, total cholesterol, and creatinine level at baseline), and Comorbidity-related variables (diabetic retinopathy, diabetic nephropathy, cardiovascular disease, hypertension at baseline, and anemia at baseline) were the study's independent variables.

## Operational definition

**Diabetic neuropathy**: After ruling out other potential explanations, the hospital doctors considered the existence of symptoms and/or indications of peripheral nerve damage among diabetic patients using EMG (Electromyography), nerve conduction velocity (NCV) tests to confirmed diabetic neuropathy [32].

**Time to diabetic neuropathy**: the time between newly diagnoses of type 2 diabetic Mellitus until the first episode of diabetic neuropathy (in months).

**Event**: Diabetic neuropathy.

**Censored**: was considered, be lost to follow-up, died, transfer out before developing the event, or be event-free at the end of the study.

**Hypertension**: is defined as systolic blood pressure (SBP)> 130mmHg and/or diastolic blood pressure > 8ommHg or current use of anti-hypertensive medication [33].

**Anemia**: A patient's hemoglobin level below 12.0 g/dl for females and below 13 g/dl for males was classified as anemia [34].

**Cardiovascular disease**: This study was considered cardiac heart disease, stroke, or peripheral arterial disease diagnosed by physicians based on the clinical assessments and confirmed using the diagnostic test [35].

**High blood glucose levels**: fasting plasma glucose levels 126mg/dl or more, or random plasma glucose or 2-hour post-load glucose levels more than 200 mg/dl [36].

## Data collection tools and procedure

The data was collected using a structured English version data abstraction checklist which was prepared by reviewing different literatures [20, 21, 26, 27]. The actual data from both DM patient's registration book and medical history registration card. Two BSc nurses as data collector and one public health professional as supervisor was recruited for each selected hospital.

## Data quality assurance

Prior to data collection, one day training was given to supervisors and data collectors on objective, data collection procedure, and data collection process and extraction checklist of the study. During data collection, daily monitoring was carried out by supervisors and principal investigator. The consistency and completeness of data were checked by the principal investigator and supervisors on daily basis.

## Data processing and analysis

The data were checked for completeness, coded, and entered into Epi-data version 3.1; and exported to Stata/SE 14.0 for data cleaning and analysis. Continuous data were reported with a mean (standard deviation) and median (interquartile range). The data with categorical nature

was described with frequency and proportion. The outcomes of study participants were dichotomized into (code '1') as an event (developing diabetic neuropathy) and (code '0') as a censor. Some continuous variables were categorized for ease of analysis and otherwise used as continuous. The variance inflation factor (VIF) and correlation matrix were used to assess multi-collinearity.

The Kaplan Meier survival curve was used to estimate survival time, and a log-rank test was used to compare the survival curves of categorical variables. The necessary assumption of the Cox-proportional hazard regression model was checked using the Schoenfeld residual test, the graphical methods, and the presence of a time-dependent covariate. The overall model adequacy and fineness were assessed using the Cox-Snell residuals and global fit test, respectively. The Log likelihood ratio was used to select the final variables of the model and also bi-variable Cox-regression was computed for each predictor variable and a P-value of <0.25 was used as a cut-off point to enter variables to multi-variable Cox-regression. The variables were selected through backward stepwise procedures. The confounding effect was minimized using proper inclusion and exclusion criteria and a multi-variable analysis. Since, the median survival time of adult type two diabetic patients was undetermined; because the largest observed analysis time was censored and the survival curve does not drop below 0.5. As the result, in this study, the restricted mean survival time of adult type two diabetic patients was estimated.

Restricted mean survival time (RMST) is suggested as a novel alternative measure in survival analyses and may be useful when proportional hazards assumption cannot be made or when event rate is low. It is defined as the area under the survival curve up to a specific time point and is generally more reliably estimable than mean or median survival times. In the case of crossing survival curves, the efficacy of an intervention may be demonstrated by showing a difference in RMST between two curves although the log-rank test may fail to detect differences. The result of the final model was expressed in terms of adjusted hazard ratio (AHR) with 95% confidence intervals. The significant association was declared with a p-value less than 0.05 in a multivariable Cox regression model. Finally, the result of this study is presented with tables, graphs, or text narrations.

## Ethical consideration

The ethical approval letter was obtained from Debre Markos University College of Health Sciences Research and Ethical Review Committee (Ref.No/ HSR /R/C/Ser/ PG/co/ 106/ 11/ 14). As the study was conducted through a review of medical records, the individual patient was not subject to harm and the official letter of co-operation to Felege Hiwot specialized hospital, Debre Tabor Specialized hospital and Debre Markos specialized hospital was taken from Debre Markos University. Permission was taken from each study hospital managers and medical ward unit leaders. According to the research review committee of College of Health Sciences, written consent was not required as confidentiality and anonymity were strictly maintained. To keep the confidentiality, name and other identifiers of patients and health care professionals were not recorded on the data extraction format. Confidentiality was maintained through anonymity and privacy measures were taken to preserve the right of the participants throughout the research work including publication. This study was conducted in accordance with the Declaration of Helsinki.

## Results

### Socio-demographic characteristics of study participants

A total of 669 charts of adult type 2 DM patients who were enrolled from March 1, 2007 to February 28, 2012 of study area hospitals were reviewed and included in the final analysis.

**Table 1. Socio-demographic characteristics of type 2 DM patients at Amhara region Comprehensive Specialized Hospital, Northwest Ethiopia (n = 669).**

| Variables | Category | Status at last contact | | Total (%) |
|---|---|---|---|---|
| | | Diabetic neuropathy (%) | Censored (%) | |
| Sex | Male | 81 (20.4) | 315 (79.6) | 396 (59.19) |
| | Female | 57 (20.9) | 216 (79.1) | 273 (40.81) |
| Age in years | ≤ 40 | 7 (9.3) | 68 (90.7) | 75 (11.21) |
| | 41–60 | 58 (15.2) | 324 (84.8) | 382 (57.10) |
| | > 60 | 73 (34.4) | 139 (65.6) | 212 (31.69) |
| Residence | Urban | 99 (18.9) | 425 (81.1) | 524 (78.33) |
| | Rural | 39 (26.9) | 106 (73.1) | 145 (21.67) |

About three hundred ninety-six (59.19%) of the study participants were males. More than three-fourth 524 (78.33%) participants were urban. The mean age of study participants at the time of diagnosis of T2DM was 54.8(±10.9) years (Table 1).

### Clinical and comorbidity characteristics of the study participants

Nearly three-fourths (74.44%) of study participants were taking an oral medication, and more than one-third (37.07%) of the individuals had hypertension at the time of the study. One hundred twenty-six (18.83 percent) of the study participants had diabetic retinopathy, and nearly one-fifth (19.58 percent) had anemia. One hundred nine (16.29%) of the study participants had diabetic nephropathy at the time of their diagnosis, and one hundred one (15.55%) had cardiovascular disease (Table 2).

### Biochemical characteristics of the study participants

In this study, the mean fasting blood sugar level was 200±60 mg/dl. Of the total participants, three hundred thirty-two (49.63%) had LDL levels≥ 100mg/dl, three hundred eighty-three

**Table 2. Clinical and comorbidity characteristics of type 2 DM patients at Amhara region Comprehensive Specialized Hospital, Northwest Ethiopia (n = 669).**

| Variables | Category | Status at last contact | | Total (%) |
|---|---|---|---|---|
| | | Diabeticneuropathy (%) (%) | Censored (%) | |
| Family history of DM | Yes | 47 (19.7) | 192 (80.3) | 239 (35.72) |
| | No | 91 (21.2) | 339 (78.8) | 430 (64.28) |
| Type of treatment | Oral | 90 (18.1) | 408 (81.9) | 498 (74.44) |
| | Insulin | 34 (30.3) | 78 (69.6) | 112 (16.74) |
| | Mixed | 14 (23.7) | 45 (76.3) | 59 (8.82) |
| Family history of complication of DM | Yes | 16 (20.8) | 61 (79.2) | 77 (11.51) |
| | No | 122 (20.6) | 470 (79.4) | 592 (88.49) |
| Diabetic retinopathy | Yes | 70 (55.5) | 56 (44.5) | 126 (18.83) |
| | No | 68 (12.5) | 475 (87.5) | 543 (81.17) |
| Diabetic nephropathy | Yes | 26 (23.9) | 83 (76.1) | 109 (16.29) |
| | No | 112 (20) | 448 (80) | 560 (83.71) |
| Cardiovascular disease | Yes | 32 (30.8) | 72 (69.2) | 104 (15.55) |
| | No | 106 (18.8) | 459 (81.2) | 565(84.45) |
| Hypertension | Yes | 97 (39) | 152 (61) | 249 (37.07) |
| | No | 41 (9.8) | 379 (89.2) | 420 (62.93) |
| Anemia | Yes | 79 (60.3) | 52 (39.7) | 131 (19.58) |
| | No | 59 (11) | 479 (89) | 538 (80.42) |

**Table 3. Biochemical characteristics of type 2 DM patients at Amhara region Comprehensive Specialized Hospital, Northwest Ethiopia (n = 669).**

| Variables | Category | Status at last contact (%) | | Total (%) |
|---|---|---|---|---|
| | | Diabetic neuropathy (%) | Censored (%) | |
| FBS | ≤ 200mg/dl | 38 (8.9) | 390 (91.1) | 428 (63.98) |
| | > 200mg/dl | 100 (41.5) | 141 (58.5) | 241 (36.02) |
| LDL | < 100mg/dl | 45 (14.3) | 292 (86.7) | 337 (50.37) |
| | ≥ 100mg/dl | 93 (28) | 239 (72) | 332 (49.63) |
| HDL | < 40mg/dl | 37 (30.3) | 85 (69.7) | 122 (18.24) |
| | ≥ 40mg/dl | 101 (18.5) | 446 (81.5) | 547 (81.76) |
| TG | < 150mg/dl | 28 (9.5) | 258 (90.5) | 286 (42.75) |
| | ≥ 150mg/dl | 110 (29.4) | 273 (70.6) | 383 (57.25) |
| CR | ≤ 1.1mg/dl | 114 (21.6) | 413 (78.4) | 527 (78.77 |
| | > 1.1mg/dl | 24 (17) | 118 (83) | 142 (21.23) |
| TC | < 200mg/dl | 31 (10.5) | 265 (89.5) | 296 (44.25) |
| | ≥ 200mg/dl | 107 (28.7) | 266 (71.3) | 373 (55.75) |
| Proteinuria | Positive | 40 (27.6) | 105 (72.4) | 145 (21.67) |
| | Negative | 98 (18.7) | 426 (81.3) | 524 (78.33) |
| Albuminuria | Positive | 34 (31.2) | 75 (68.8) | 109 (16.29) |
| | Negative | 104 (18.6) | 456 (81.4) | 560 (83.71) |

**Abbreviations:** FBS, Fasting Blood Sugar, HDL, High-Density Lipoprotein, LDL, Low-Density Lipoprotein, CR, creatinine, TG, Triglyceride, TC, Total Cholesterol, mg/dl, Milligram Deciliter

(57.25%) had triglyceride levels≥ 150mg/dl. and one hundred twenty-two (18.24%) had high-density lipoprotein level < 40mg/dl. Of the total respondents, one hundred forty-five (21.67%), one hundred nine (16.29%) were develop proteinuria and albuminuria respectively (Table 3).

## Diabetic neuropathy adult type two diabetes mellitus patients

During the follow-up period of the study, one hundred thirty-eight (20.63%) T2DM patients developed diabetic neuropathy, and five hundred thirty-one (79.37%) were censored (Fig 1).

## Survival experiences of type 2 DM patients

Statistical difference in survival time between different categorical variables was tested using the Log-rank test. There was a significant difference in survival experiences among type of treatment, diabetic retinopathy, diabetic nephropathy, cardiovascular disease, hypertension, anemia, proteinuria, and albuminuria at a p-value < 0.05. However, there is no significant difference among categories of sex, residence family history of DM, and family history of complications of DM. The restricted mean survival time to develop diabetic neuropathy for T2DM patients who were treated with oral treatment was longer [182 months (95% CI: 176.18–188.43)] than for those treated with Insulin and mixed medication. The restricted mean survival time to develop diabetic neuropathy for T2DM patients with diabetic retinopathy was shorter [107 months (95% CI: 97.36–118.44)] than for those who had no diabetic retinopathy [193 months (95% CI: 187.34–198.39)] (Table 4).

The restricted mean survival time to develop diabetic neuropathy for T2DM patients with anemia was shorter [110 months (95% CI: 100.36–119.66)] than for those who had no anemia [196 months (95% CI: 192.37–201.15)]. The restricted mean survival time to develop diabetic

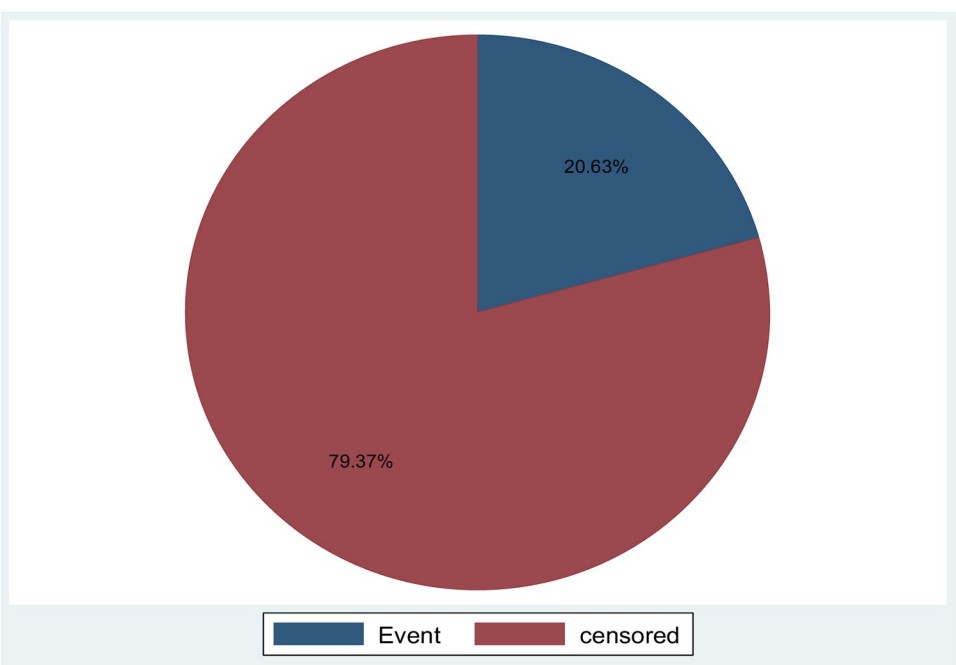

**Fig 1. Outcomes of adult type two diabetes mellitus patients Amhara region Comprehensive Specialized Hospital, Northwest Ethiopia, 2022 (n = 669).**

neuropathy for T2DM patients with Hypertension had shorter [142 months (95% CI: 131.68–152.75)] than for those who had no hypertension (Table 4).

The log-rank test shows a significant difference between insulin treatment and oral treatment in survival experience to develop diabetic neuropathy. Patients who were treated with Insulin at diagnosis had a shorter survival probability to develop diabetic neuropathy than those treated with oral medication and Mixed (p = 0.004) (Table 4). Diabetic patients who had hypertension at diagnosis had a shorter survival probability to develop diabetic neuropathy than those DM patients who had no hypertension (p = 0.001) (Table 4).

The log-rank test shows a significant difference between anemia and anemia-free T2DM patients' survival experience to develop diabetic neuropathy. Patients who had anemia at diagnosis had a shorter survival probability to develop diabetic neuropathy than those DM patients who had no anemia (p = 0.001) (Table 4). In addition, patients who had diabetic retinopathy at diagnosis had a longer survival probability than patients who had no diabetic retinopathy (p = 0.001). Furthermore, the log-rank test shows a significant difference between cardiovascular disease and cardiovascular disease-free T2DM patients' survival experience to develop diabetic neuropathy. DM patients who had cardiovascular disease at diagnosis had a shorter survival probability to develop diabetic neuropathy than those DM patients who had not a cardiovascular disease (p = 0.001) (Table 4).

## Overall survival of adult type two diabetes mellitus patients

In this study, 669 Adult type two diabetes mellitus patients were followed for a minimum of 13 months and a maximum of 215 months. The median follow-up time and the restricted mean survival time of this study were 124 [IQR: 85.9, 148.5] and 179 [95% CI: 173.7, 185.1] months, respectively. The overall estimated survival rate of Adult type two diabetes mellitus patients was 68.45% at 215 months of follow-up. The estimated cumulative survival probability of

**Table 4. Restricted mean survival time of type 2 DM patients at Amhara region Comprehensive Specialized Hospital, Northwest Ethiopia (n = 669).**

| Variables | Category | RMST, in months (95% CI) | Log-rank test | |
|---|---|---|---|---|
| | | | X2 | p-value |
| Sex | Male | 179.6 [172.24, 187.12] | 1.08 | 0.701 |
| | Female | 177.5 [169.65, 185.40] | | |
| Residence | Urban | 156.1 [152.09, 160.25] | 1.96 | 0.162 |
| | Rural | 172.2 [160.32, 184.11] | | |
| Family history of DM | Yes | 182.7 [175.31, 190.21] | 1.32 | 0.250 |
| | No | 176.2 [168.43, 184.13] | | |
| Type of treatment | Oral | 182.3 [176.18, 188.43] | 15.59 | 0.004* |
| | Insulin | 158.9 [143.47, 174.34] | | |
| | Mixed | 147.5 [134.42, 160.64] | | |
| Family history of complications of DM | Yes | 155.5 [145.44, 165.58] | 1.21 | 0.863 |
| | No | 178.8 [172.62, 185.06] | | |
| Diabetic retinopathy | Yes | 107.9 [97.36, 118.44] | 176.57 | < 0.001* |
| | No | 193.1 [187.34, 198.39] | | |
| Diabetic nephropathy | Yes | 142.3 [187.67, 198.67] | 4.24 | 0.039* |
| | No | 181.3 [175.33, 187.26] | | |
| Cardiovascular disease | Yes | 136.1 [124.60, 147.77] | 16.03 | 0.001* |
| | No | 184.7 [179.59, 189.93] | | |
| Hypertension | Yes | 142.2 [131.68, 152.75] | 131.96 | < 0.001* |
| | No | 195.8 [190.33, 201.27] | | |
| Anemia | Yes | 110.0 [100.36, 119.66] | 191.32 | < 0.001* |
| | No | 196.7 [192.37, 201.15] | | |
| Proteinuria | Positive | 140.3 [131.52, 149.26] | 10.95 | 0.009* |
| | Negative | 183.4 [177.40, 189.46] | | |
| Albuminuria | Positive | 132.8 [122.28,143.38] | 18.61 | < 0.001* |
| | Negative | 183.6 [177.76,189.51] | | |

* indicates the significantly associated categorical variables at p < 0.05 in the Log-rank test for equality of survivor functions with a 95% confidence level.

T2DM Patients was 98.8%, 93.4%, 81.98%, and 68.76%, at the end of the two years, five years, ten years, and fifteen years respectively. About half (50.7%) of diabetes mellitus developed diabetic neuropathy within the 6 years of follow-up. The incidence rate of DN among patients with DM in this study was 2.14 per 100 person-years for 6432.88 person-years of observation. While the median survival time was undetermined because the largest observed analysis time was censored, the survivor function does not go to zero; in this case, the restricted mean is the best estimate of survival time (Fig 2).

## Predictors of time to diabetic neuropathy among Type 2 diabetic patients

Variables like age, residence, type of treatment, diabetic retinopathy, cardiovascular disease, hypertension, anemia, fasting blood sugar, low-density lipoprotein level, high-density lipoprotein level, triglyceride level, total cholesterol level, proteinuria, and albuminuria were having a *p*-value < 0.20 in the bi-variable analysis and candidate for multivariable Cox regression analysis.

The necessary assumption of the Cox-proportional hazard regression model was checked using the Schoenfeld residual test (0.057 to 0.886, with a global *P*-value of 0.726), the graphical methods, and the presence of a time-dependent covariate. The Cox Snell residual plot showed

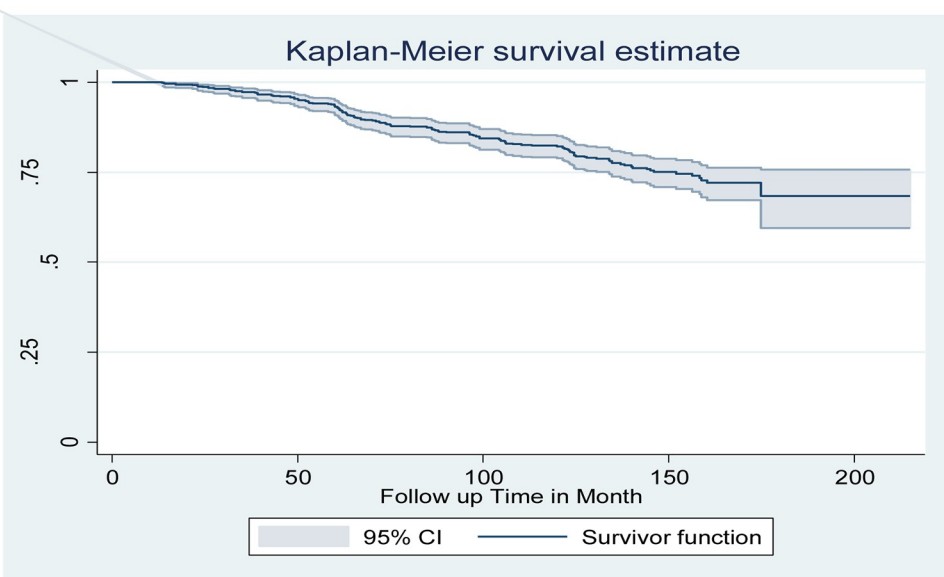

**Fig 2. Overall the Kaplan-Meier survival curves comparing survival time of adult type two DM patients in Amhara region Comprehensive Specialized Hospital, Northwest Ethiopia, 2022 (n = 669).**

the goodness of fitness of the model was satisfied because the cumulative hazard plot follows 45 degrees or a straight line through the origin with slope one (Fig 3).

As shown in Fig 3 Nelson-Aalen cumulative hazard functions follows the 45-degree line very closely except for some large values of time, so that the final model fits the data well. After

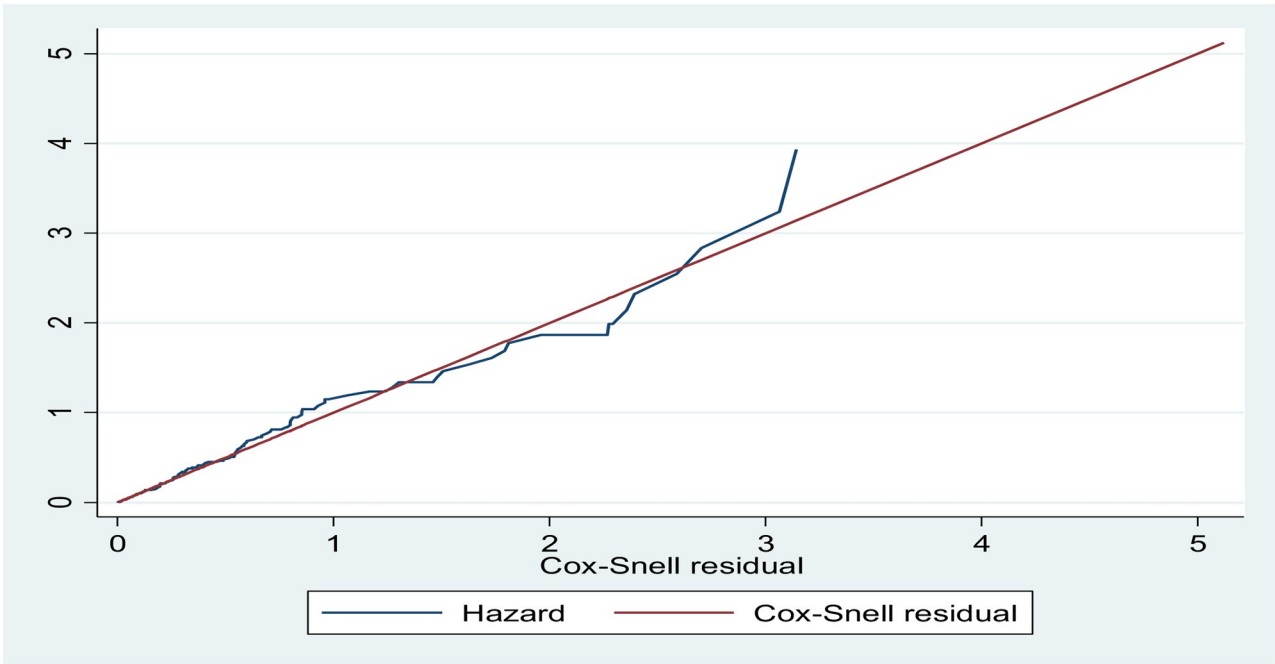

**Fig 3. Cox-Snell residual graph, based on the Kaplan–Meier estimated survivor function, to test the overall adequacy of the cox proportional hazard model of time to diabetic neuropathy and its predictors of type 2 DM patients at Amhara region Comprehensive Specialized Hospital, Northwest Ethiopia (n = 669).**

controlling the effect of other variables, the multivariable Cox regression analysis, baseline age in years, diabetic retinopathy, hypertension, anemia, and baseline fasting blood sugar level were predictors of the time to development of diabetic neuropathy.

The risk of developing diabetic neuropathy among type two diabetes mellitus patients with an age > 60 years was 2.9 times higher as compared to age ≤ 40 years [AHR: 2.93 (95% CI: 1.29–6.66)]. The risk of developing diabetic neuropathy among diabetic retinopathy type two diabetes mellitus patients was 2.7 times higher than their counterparts [AHR: 2.76 (95% CI: 1.84–4.16)]. The risk of developing diabetic neuropathy among hypertensive type two diabetes mellitus patients was 3.2 times higher than that of non-hypertensive patients [AHR: 3.22(95% CI: 2.10–4.93)]. The risk of developing diabetic neuropathy among anemic type two diabetes mellitus patients was 3.6 times higher than that of non- anemic diabetes mellitus patients [AHR: 3.62 (95% CI: 2.46–5.33)]. Moreover, risk of developing diabetic neuropathy among type two diabetic patients with baseline fasting blood sugar level of above 200 mg/dl were 2.56 times higher than their counter parts [AHR: 2.56 (95% CI: 1.68–3.92)] (Table 5).

## Discussion

This study aimed to assess the time to diabetic neuropathy and predictors among adult type two diabetic patients who were free at baseline in the Amhara region compressive specialized hospitals. The present study revealed that restricted mean time to develop diabetic neuropathy was 179.5 [95% CI: 173.7, 185.1] months which, means that the average time of type 2 Diabetes mellitus patients living without developing diabetic neuropathy was 179.5 months. During the follow-up, 20.6% of T2DM patients developed diabetic neuropathy. About half (50.7%)of the event of interest (Diabetic neuropathy) occurs within the 6 years of follow-up, making the overall incidence rate 2.14 cases per 100 person-year of follow-up. This finding was in line with that of the study conducted in the university of Gondar compressive specialized hospital, in Ethiopia (2.01 cases per 100 person-year of follow-up) [27], and in the United States of America (26.9 per 1000 person-years) [37]. This might be because diabetic patients receive nearly identical medical care. The follow-up time for the study conducted in the United States of America and this study was nearly same, at over 15 years and 16 years, respectively. This may account for the similar incidence of diabetic neuropathy. However, the finding of the current study was lower than the findings of study done in Karachi-Pakistan (106.2 per 1000 person-years) [38]. This finding was higher than that of the study conducted at the University of Gondar compressive specialized hospital (44% by the end of the follow-up period) [27]. This difference might be due to the follow-up period included in the study; our study included 16 years T2DM patients' data whereas the study conducted in the University of Gondar compressive specialized hospital study used 19 years T2DM patients' data.

The hazard of developing Diabetic neuropathy among type two diabetic patients with retinopathy was higher than in type two DM patients without diabetic retinopathy. This finding is consistent with the findings of studies conducted in Sri Lanka [21], Netherlands [39], Jordan [40], Singapore [41], and china [28]. The clinical stages of diabetic retinopathy, the ongoing disturbance of the neurovascular unit, and the loss of auto-regulation may all be to blame for this. Therefore, changes to the retinal neurovascular unit in diabetes are a component of a wider range of impacts on the nervous system that also includes cognitive impairment and sensory and autonomic neuropathies [42], and also some evidence shows that oxidative stress, the formation of advanced glycation end products (AGEs), and hemodynamic changes have been specifically implicated in neuro-degeneration during Diabetic retinopathy [43].

This study also found that the hazard of developing Diabetic neuropathy among T2DM patients with hypertension was higher than their counter parts. This finding is consistent with

**Table 5. Result of the bi-variable and multi-variable cox regression analysis of adult T2DM in Amhara region compressive specialized hospital Northwest Ethiopia, 2022 (n = 669).**

| Variables | Category | Diabetic neuropathy N (%) | Censored N (%) | Bi-variable CHR (95% CI) | Multi-variable CHR (95% CI) |
|---|---|---|---|---|---|
| Age at diagnosis | ≤ 40 | 7 (1.04) | 68(10.16) | 1 | 1 |
| | 41–60 | 58 (8.67) | 324 (48.43) | 1.74(0.7–3.8) | 1.59(0.7–3.5) |
| | > 60 | 73 (10.9) | 139 (20.77) | 5.31(2.4–11.5) | 5.31(2.4–11.5) |
| Residence | Urban | 99 (14.79) | 425 (63.53) | 0.76 (0.5–1.1) | 1.16(0.7–1.7) |
| | Rural | 39 (5.83) | 106 (15.85) | 1 | 1 |
| Type of treatment | Oral | 90 (13.48) | 408 (60.98) | 1 | 1 |
| | Insulin | 34 (5.08) | 78 (11.65) | 2.15(1.4–3.1) | 1.34(0.8–2.0) |
| | Mixed | 14 (2.09) | 45 (6.72) | 1.49(.8–2.6) | 1.34 (0.7–2.4) |
| Diabetic retinopathy | Yes | 70 (10.46) | 56 (8.37) | 7.16(5.1–10.0) | 2.76 (1.8–4.1) |
| | No | 68 (10.16) | 475 (71.01) | 1 | 1 |
| Diabetic nephropathy | Yes | 26 (3.88) | 83 (12.4) | 1.56(1.0–2.3) | 1.28(0.7–2.1) |
| | No | 112 (16.74) | 448 (66.98) | 1 | 1 |
| Cardiovascular disease | Yes | 32 (4.70) | 72 (10.76) | 2.20(1.4–3.2) | 1.03(0.6–1.5) |
| | No | 106 (15.84) | 459 (68.70) | 1 | 1 |
| Hypertension | Yes | 97 (14.49) | 152 (22.72) | 6.70(4.6–9.7) | 3.22(2.1–4.9) |
| | No | 41 (6.13) | 379 (56.66) | 1 | 1 |
| Anemia | Yes | 79 (11.80) | 52 (7.77) | 7.64(5.4–10.7) | 3.62(2.4–5.3) |
| | No | 59 (8.81) | 479 (72.62) | 1 | 1 |
| FBS | FBS≤200 mg/dl | 38 (5.68) | 390 (58.30) | 1 | 1 |
| | FBS > 200 mg/dl | 100 (14.94) | 141 (21.08) | 5.73(3.9–8.3) | 2.56 (1.6–3.9) |
| HDL | HDL < 40 mg/dl | 37 (5.53) | 85 (12.70) | 2.32(1.5–3.3) | 1.18(0.7–1.8) |
| | HDL ≥40 mg/dl | 101(15.10) | 446 (66.67) | 1 | 1 |
| LDL | LDL < 100 mg/dl | 45 (6.72) | 292 (43.64) | 1 | 1 |
| | LDL ≥ 100 mg/dl | 93 (13.90) | 239 (35.74) | 2.39(1.6–3.4) | 1.09(0.7–1.6) |
| TG | TG < 150 mg/dl | 28 (4.20) | 258 (38.56) | 1 | 1 |
| | TG ≥ 150 mg/dl | 110 (16.44) | 273 (40.8) | 3.52(2.3–5.3) | 0.92 (0.5–1.5) |
| TC | TC < 200 mg/dl | 31 (4.63) | 265 (39.61) | 1 | 1 |
| | TC ≥200 mg/dl | 107 (15.99) | 266 (39.77) | 3.23 (2.1–4.8) | 1.37 (0.8–2.2) |
| Protein urea | Positive | 40 (5.98) | 105 (15.69) | 1.84 (1.2–2.6) | 0.90 (0.5–1.5) |
| | Negative | 98 (14.64) | 426 (63.69) | 1 | 1 |
| Albumin urea | Positive | 34 (5.08) | 75 (11.21) | 2.29(1.5–3.3) | 1.15(.6–2.0) |
| | Negative | 104 (15.54) | 456 (68.17) | 1 | 1 |

**Note**:

* indicates the variables significantly associated with the outcome variable at < **0.05** in multivariable analysis with a 95% confidence level

previous studies conducted in Gamo Gofa, Ethiopia [26], Tanzania [21], Burkina Faso [44], and Netherland [39]. This might be due to inflammation, oxidative stress, activation of the immune system, thickening of the blood vessels which results reduce blood flow to the nerve, conduction slowing, and axonal atrophy [45]. Nonetheless, the finding of this study was inconsistency with the study conducted in Brazil [46]. This might be due to difference in study design, health-seeking behavior, socio-economic and socio-demographic characteristics and sample size; the former study was cross-sectional while this study was retrospective follow up. In addition, compared to T2DM patients under the age of 40, patients with T2DM who were older than 60 had a higher chance of developing diabetic neuropathy.

This finding is consistent with previous studies conducted in Jima compressive specialized hospital [19], UGCSH [27], Tanzania [47], Egypt [48], Uganda [49], India [50], Singapore [41], and China [48]. This could be the result of aging brought on by fat accumulation, oxidative stress, activation of counter-regulatory signaling pathways, and mitochondrial malfunction, all of which contribute to increased inflammation and damage to the peripheral nerves [51]. T2DM patients who had baseline fasting blood sugar levels above 200 mg/dl had a 2.5-fold increased chance of developing diabetic neuropathy compared to those who had baseline fasting blood sugar levels below 200 mg/dl. This is consistent with a case-control study conducted in Gamo Gofa, Ethiopia [26]. The possible explanation is that uncontrolled high blood sugar damages nerves and interferes with their ability to send signals, leading to diabetic neuropathy [52]. This association holds true but is lesser than a study conducted in Egypt [53] and Iran [54]. This might due to difference in study design, health-seeking behavior, and the socio-demographic characteristics.

Furthermore, this study identified that T2DM patients with anemia significantly increased the hazard of diabetic neuropathy development as compared to non-anemic patients. This finding is consistent with the previous study conducted at University of Gondar specialized hospital [27] and China [55]. The possible explanation might be that anemia is considered to be associated with oxidative stress [56] which is also an important mechanism of diabetic neuropathy.

## Limitations of the study

Despite thorough efforts to mitigate the study's potential flaws, the current study have some limitations; due to the retrospective nature of our study, this study was unable to investigate the role of some socio-demographic and behavioral factors, such as level of education, alcohol consumption, smoking status, income level, and exercise in diabetic neuropathy due to a lack of data. In addition, due to incomplete data, in this study, broader risk factors such as body mass index were unable to consider.

## Conclusion

The restricted mean survival time of this study was 179.4 months which is a relatively short time and the risk of occurrence of diabetic neuropathy among type two diabetes mellitus patients was high in the early period and the incidence density rate of diabetic neuropathy was relatively high.

Variables like age in years, diabetic retinopathy, anemia, baseline fasting blood sugar level> 200 mg/dl, and hypertension were significantly associated with diabetic neuropathy. Therefore, Amhara Region Comprehensive Specialized Hospitals need to strengthen the follow-up T2DM patients with Hypertension, high fasting blood sugar, anemia, old age, and patients who had diabetic retinopathy to reduce the risk of diabetic neuropathy.

## Acknowledgments

Authors would like to thank the Debre Markos University College of Health Sciences for providing opportunity to conduct this research work. In addition, our gratitude goes to all hospitals under the region for their cooperation and providing necessary information. Also, we are grateful to acknowledge our study participants for providing the necessary information and the data collectors for collecting the data carefully.

## Author Contributions

**Conceptualization:** Sharie Tantigegn, Atsede Alle Ewunetie, Moges Agazhe, Abiot Aschale, Muluye Gebrie, Gedefaw Diress, Bekalu Endalew Alamneh.

**Data curation:** Sharie Tantigegn, Atsede Alle Ewunetie, Moges Agazhe, Abiot Aschale, Muluye Gebrie, Gedefaw Diress, Bekalu Endalew Alamneh.

**Formal analysis:** Sharie Tantigegn, Atsede Alle Ewunetie, Abiot Aschale, Muluye Gebrie, Gedefaw Diress, Bekalu Endalew Alamneh.

**Funding acquisition:** Sharie Tantigegn, Atsede Alle Ewunetie, Moges Agazhe, Abiot Aschale, Muluye Gebrie, Gedefaw Diress, Bekalu Endalew Alamneh.

**Investigation:** Sharie Tantigegn, Atsede Alle Ewunetie, Moges Agazhe, Abiot Aschale, Muluye Gebrie, Gedefaw Diress.

**Methodology:** Sharie Tantigegn, Atsede Alle Ewunetie, Moges Agazhe, Abiot Aschale, Muluye Gebrie, Gedefaw Diress, Bekalu Endalew Alamneh.

**Project administration:** Sharie Tantigegn, Atsede Alle Ewunetie, Moges Agazhe, Abiot Aschale, Muluye Gebrie.

**Resources:** Sharie Tantigegn, Atsede Alle Ewunetie, Abiot Aschale, Muluye Gebrie, Gedefaw Diress, Bekalu Endalew Alamneh.

**Software:** Sharie Tantigegn, Atsede Alle Ewunetie, Moges Agazhe, Abiot Aschale, Muluye Gebrie, Gedefaw Diress, Bekalu Endalew Alamneh.

**Supervision:** Atsede Alle Ewunetie, Moges Agazhe, Muluye Gebrie, Gedefaw Diress.

**Validation:** Sharie Tantigegn, Moges Agazhe, Abiot Aschale, Muluye Gebrie, Gedefaw Diress, Bekalu Endalew Alamneh.

**Visualization:** Sharie Tantigegn, Atsede Alle Ewunetie, Abiot Aschale, Muluye Gebrie, Gedefaw Diress, Bekalu Endalew Alamneh.

**Writing – original draft:** Sharie Tantigegn, Moges Agazhe, Bekalu Endalew Alamneh.

**Writing – review & editing:** Sharie Tantigegn, Muluye Gebrie, Gedefaw Diress.

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
