## [Decision Letter · Decision Letter 0]

21 Nov 2022

PONE-D-22-25165

Time to Diabetic Neuropathy and Its Predictors among Adult Type 2 Diabetes Mellitus Patients in Amhara Regional State Comprehensive Specialized Hospitals, Northwest Ethiopia, 2022: A Retrospective Follow up Study

PLOS ONE

Dear Dr. Alamneh,

Thank you for submitting your manuscript to PLOS ONE. After careful consideration, we feel that it has merit but does not fully meet PLOS ONE’s publication criteria as it currently stands. Therefore, we invite you to submit a revised version of the manuscript that addresses the points raised during the review process.

We look forward to receiving your revised manuscript.

Kind regards,

James Nyirenda

Academic Editor

PLOS ONE

Journal Requirements:

2. In the ethics statement in the manuscript and in the online submission form, please provide additional information about the patient records/samples used in your retrospective study. Specifically, please ensure that you have discussed whether all data/samples were fully anonymized before you accessed them and/or whether the IRB or ethics committee waived the requirement for informed consent.

Reviewers' comments:

Reviewer's Responses to Questions

**Comments to the Author**

1. Is the manuscript technically sound, and do the data support the conclusions?

Reviewer #1: Yes

Reviewer #2: Yes

Reviewer #3: Partly

2. Has the statistical analysis been performed appropriately and rigorously? 

Reviewer #1: Yes

Reviewer #2: Yes

Reviewer #3: No

3. Have the authors made all data underlying the findings in their manuscript fully available?

Reviewer #1: Yes

Reviewer #2: Yes

Reviewer #3: No

4. Is the manuscript presented in an intelligible fashion and written in standard English?

Reviewer #1: Yes

Reviewer #2: Yes

Reviewer #3: No

5. Review Comments to the Author

Reviewer #1: Abstract:

Abstract is written well with clear background, objectives, methods, and conclusion. It reflected the content of the manuscript.

Introduction:

The background of the study was presented clearly and supported with relevant recent citations. The problem statement and rationale of the study were clear. The length of the introduction was appropriate compared to the length of the manuscript.

Methods:

The study design matches the study objectives. However, the time frame for participant enrolment and time frame for follow up need to be clearly stated.

The type of oral antidiabetic therapy used to treat the patients should be specified bearing in mind that some antidiabetic drugs (e.g., metformin) promote development of neuropathy on chronic use.

It is not clear why co-morbidities and biochemical markers such as blood glucose, cholesterol, and creatinine were only measured at baseline. Please provide justification for this!

There was no mention of measurement of long-term glycemic control during the follow up period for a project that lasted more than 10 years!!

Results:

Table and figure titles should be separated from main text.

In table 3, what is the basis for categorizing FBS based on a value of 200 mg/dl, cholesterol 200 mg/dl, and LDL 100 mg/dl?

In table 3 still, total number with “proteinuria” was 145, while total number with “albuminuria” was 109. How did you distinguish between proteinuria and albuminuria?

In table 4, explain RSMT (restricted mean survival time) as a footnote.

Discussion:

Interpretations of the findings and conclusions to be drawn from the data presented should consider the concerns raised in this review.

References:

Your list of references did not comply with PLOS One specifications (Vancouver style) and there are obvious errors in many of the entries.

Reviewer #2: Overall interesting study. Time to Diabetic Neuropathy and Its Predictors among Adult Type 2 Diabetes Mellitus Patients in Amhara Regional State Comprehensive Specialized Hospitals, Northwest Ethiopia, 2022: A Retrospective Follow up Study. However, the manuscript does not provide a clear overview of their work.

• Please make sure that the structure for citing published literature in the text, as well as the style of references in the References section, are consistent with the journal's style (see Instructions to Authors).

• English language needs revision for style and syntax.

• Abstract must be rewritten. I suggest focusing the abstract on your study and your results.

• Include more characteristics of participants.

• Please specify inclusion/exclusion criteria. The experimental protocol is not clear.

• Please add the originality of the study and add hypothesis at the end of the introduction section. Be please be more specific.

• Did authors perform other statistical analysis? Further details on statistical analysis are needed. Please be more specific.

• Figure 2 (axis) not clear. Tables must be more representative

• Please discuss the results of the study in relation to the previous studies.

Reviewer #3: The study by Alamneh et al., titled ‘Time to diabetic neuropathy and its predictors among adult type 2 diabetes mellitus patients in Amhara Regional State Comprehensiv Specialized Hospitals, Northwest Ethiopia, 2022: A retrospective follow up study’ aimed to evaluate the time to diabetic neuropathy and its determinants. The authors analyzed data of 669 newly diagnosed with T2D during the 5 year enrollment period and followed up for a median 125 month. They found the overall incidence rate of diabetic neuropathy as 2.1 cases per 100 persons-year. They identified age (>60 years), diabetic retinopathy, anemia, hypertension, and hyperglycemia (FPG>200 mg/dl) as predictors of incident diabetic neuropathy.

Major comments:

1. The manuscript should be written in a more fluent fashion and be more concise and comprehensive. There are too unnecessary repetitions, many grammatical errors, irrelevant decimals (for example instead of 179,45 months, an expression of 179 months may be easier and better for the reader). Overall, it needs an extensive editing

2. It seems that the definition of diabetic retinopathy is based on subjective, but not objective criteria (such as EMG).

3. What kind of anemia? Iron deficiency? Vitamin B 12 deficiency? Folate deficiency? Chronic disease anemia? Thalassemia trait? Sickle cell trait?

4. Were vitamin B12 levels measured?

5. What kind of oral treatments were used? Metformin? Sulfonylureas? Pioglitazone? SGLT2 inh? DPPIV inh.?

6. Was not Injectable GLP-1 analogues available during the study period?

7. What kind of insulins were used? Basal, basal-bolus, mixed?

8. Did the authors monitor glycemic status of the patients? For example, glycemic fluctuations are one of the major contributors to the diabetic neuropathy development. Was there hypoglycemia episodes during the study, did the authors consider this parameter when performing the analysis?

6. PLOS authors have the option to publish the peer review history of their article (what does this mean?). If published, this will include your full peer review and any attached files.

Reviewer #1: **Yes: **Christian Chinyere Ezeala

Reviewer #2: No

Reviewer #3: No

---

## [Author Response · Author response to Decision Letter 0]

19 Feb 2023

Pont by point response 

Reviewers' comments:

Reviewer's Responses to Questions

Comments to the Author

1. Is the manuscript technically sound, and do the data support the conclusions? The manuscript must describe a technically sound piece of scientific research with data that supports the conclusions. Experiments must have been conducted rigorously, with appropriate controls, replication, and sample sizes. The conclusions must be drawn appropriately based on the data presented.

Reviewer #1: Yes

Reviewer #2: Yes

Reviewer #3: Partly

Authors’ response: we have appreciated the insight of the reviewers and amended the revised manuscript as per standard of writing a conclusion. 

 2. Has the statistical analysis been performed appropriately and rigorously?

Reviewer #1: Yes

Reviewer #2: Yes

Reviewer #3: No

Authors’ response: Thank you for the comments. We have checked the analysis of this study thoroughly and it is appropriate for the collected data. In addition we have tried to elaborate the analysis part in the section methods part of the revised manuscript. 

3. Have the authors made all data underlying the findings in their manuscript

fully available? The PLOS Data policy requires authors to make all data underlying the findings

described in their manuscript fully available without restriction, with rare exception (please refer to the Data Availability Statement in the manuscript PDF file). The data should be provided as part of the manuscript or its supporting information, or deposited to a public repository. For example, in addition to summary statistics, the data points behind means, medians and variance

measures should be available. If there are restrictions on publicly sharing data e.g. participant privacy or use of data from a third party those must be specified.

Reviewer #1: Yes

Reviewer #2: Yes

Reviewer #3: No

Authors’ response: we made the data available as per the request and the data is fully available at the hand of the principal investigator. 

 4. Is the manuscript presented in an intelligible fashion and written in Standard English?

Reviewer #1: Yes

Reviewer #2: Yes

Reviewer #3: No

Authors’ response: We really admire the comments regarding manuscript presentation and grammar standards. We have checked the entire document carefully and amended accordingly in the revised manuscript and highlighted. 

5. Review Comments to the Author

Please use the space provided to explain your answers to the questions above.

You may also include additional comments for the author, including concerns

about dual publication, research ethics, or publication ethics. (Please upload

your review as an attachment if it exceeds 20,000 characters)

Reviewer #1: Abstract:

Abstract is written well with clear background, objectives, methods, and

conclusion. It reflected the content of the manuscript.

Authors’ response: Accepted 

Introduction: The background of the study was presented clearly and supported with relevant

recent citations. The problem statement and rationale of the study were clear. The length of the introduction was appropriate compared to the length of the manuscript.

Authors’ response: Accepted 

Methods: The study design matches the study objectives. However, the time frame for

participant enrolment and time frame for follow up need to be clearly stated. 

Authors’ response: We really noted the raised issue. Regarding time of enrollment and follow up, we have clearly stated in the revised manuscript. See line 101-102. 

The type of oral antidiabetic therapy used to treat the patients should be specified bearing in mind that some antidiabetic drugs (e.g., metformin) promote development of neuropathy on chronic use.

Authors’ response: Yes, as you stated metformin will contribute for the development of neuropathy in long time usage but the study participants of this study was treated by oral Gliben clamide and insulin. 

It is not clear why co-morbidities and biochemical markers such as blood glucose, cholesterol, and creatinine were only measured at baseline. Please provide justification for this!

Authors’ response: The objective of this study was to assess time to diabetic neuropathy and determinant factors by using retrospective study design. As we know in longitudinal study design our study participants will be selected by their exposure status and we follow until the development of the outcome. Particularly for this study both the outcome and exposure status was gained from the patients’ registration book and medical history as the study was conducted using secondary data. Therefore, our aim was knowing the exposure status at the beginning of the study (baseline biochemical markers and comorbidity, then follow them until the development of neuropathy i.e. does the risk is high in exposed group or in non-exposed). 

There was no mention of measurement of long-term glycemic control during the

follow up period for a project that lasted more than 10 years!!

 Authors’ response: We appreciate the intension but our objective was not on measuring the glycemic control rather time to diabetic neuropathy. Despite it is one factor for the development of diabetic neuropathy, we were un able to measure the long term glycemic control due to the nature of our data source i.e. secondary data and the objective of this study was not measuring it. We strongly recommend to consider this variable for the next researchers using prospective cohort study design.

Results: 

Table and figure titles should be separated from main text.

Authors’ response: we done it. See in the revised manuscript.

In table 3, what is the basis for categorizing FBS based on a value of 200 mg/dl,

cholesterol 200 mg/dl, and LDL 100 mg/dl?

Authors’ response: As usual when we categorize a certain variable there should be standard or it should be biologically plausible or there should be previous literatures. Hence we categorized the above variable based on the previous literatures and biological plausibility, if the FBS is above 200 it will lead to complication, LDL of 100mg/dl level is standard cut off point. Related Literatures are listed here 

https://www.hindawi.com/journals/jdr/2020/9562920/ Prevalence and Determinants of Peripheral Neuropathy among Type 2 Adult Diabetes Patients Attending Jimma University Medical Center, Southwest Ethiopia, 2019, an Institutional-Based Cross-Sectional Study

https://journals.plos.org/plosone/article?id=10.1371/journal.pone.0246722 Determinants of peripheral neuropathy among diabetic patients under follow-up in chronic care clinics of public hospitals at Gamo and Gofa zones, southern Ethiopia

https://www.ncbi.nlm.nih.gov/pmc/articles/PMC7569060/ Incidence of Diabetic Foot Ulcer and Its Predictors among Diabetes Mellitus Patients at Felege Hiwot Referral Hospital, Bahir Dar, Northwest Ethiopia: A Retrospective Follow-Up Study

In table 3 still, total number with “proteinuria” was 145, while total number with

“albuminuria” was 109. How did you distinguish between proteinuria and

albuminuria?

Authors’ response: As we know assessing for proteinuria, an established marker for chronic kidney disease (CKD). Higher protein levels are associated with more rapid progression of CKD to end-stage renal disease and increased risk for cardiovascular events and mortality in both the nondiabetic and diabetic populations. Monitoring proteinuria levels can also aid in evaluating response to treatment whereas Albuminuria is a very common (though not universal) finding in CKD patients; is the earliest indicator of glomerular diseases, such as diabetic- glomerulosclerosis; and is typically present even before a decrease in the glomerular filtration rate (GFR) or a rise in the serum creatinine. Hence knowing both proteinuria and albuminuria is very important. We were distinguish it through laboratorial results since our study was conducted at Health facilities having advanced laboratory setup. 

In table 4, explain RSMT (restricted mean survival time) as a footnote.

Authors’ response: really we appreciated it. In the revised manuscript, we tried to explain the RSMT very well in the analysis part of the method section. See line 185-191 of the revised manuscript. 

Discussion:

Interpretations of the findings and conclusions to be drawn from the data presented should consider the concerns raised in this review.

Authors’ response: we have done it accordingly.

References:

Your list of references did not comply with PLOS One specifications (Vancouver

style) and there are obvious errors in many of the entries.

Authors’ response: yes, we have checked it and managed the reference part.

Reviewer #2: Overall interesting study. Time to Diabetic Neuropathy and Its

Predictors among Adult Type 2 Diabetes Mellitus Patients in Amhara Regional

State Comprehensive Specialized Hospitals, Northwest Ethiopia, 2022: A

Retrospective Follow up Study. However, the manuscript does not provide a

clear overview of their work.

Authors’ response: we have tried to see it thoroughly and made it clear.

• Please make sure that the structure for citing published literature in the text,

as well as the style of references in the References section, are consistent with

the journal's style (see Instructions to Authors).

Authors’ response: we admired your comments regarding the referencing and corrected it.

• English language needs revision for style and syntax. 

Authors’ response: we addressed the language related issue. See the revised manuscript. 

• Abstract must be rewritten. I suggest focusing the abstract on your study and your results. Include more characteristics of participants.

Authors’ response: revised accordingly. 

• Please specify inclusion/exclusion criteria. The experimental protocol is not

clear. 

Authors’ response: we made it clear accordingly.

• Please add the originality of the study and add hypothesis at the end of the

introduction section. Be please be more specific. 

Authors’ response: We tried to add the hypothesis at the end of the introduction part of the revised manuscript.

• Did authors perform other statistical analysis? Further details on statistical

analysis are needed. Please be more specific.

Authors’ response: we have clearly explained the analysis part at the method section of the revised manuscript. See line 163-194.

• Figure 2 (axis) not clear. Tables must be more representative

Authors’ response: we have checked it and corrected it.

• Please discuss the results of the study in relation to the previous studies.

Authors’ response: Discussed accordingly.

Reviewer #3: The study by Alamneh et al., titled ‘Time to diabetic neuropathy and its predictors among adult type 2 diabetes mellitus patients in Amhara Regional State Comprehensive Specialized Hospitals, Northwest Ethiopia, 2022: A retrospective follow up study’ aimed to evaluate the time to diabetic neuropathy and its determinants. The authors analyzed data of 669 newly diagnosed with T2D during the 5 year enrollment period and followed up for a median 125 month. They found the overall incidence rate of diabetic neuropathy as 2.1 cases per 100 persons-year. They identified age (>60 years), diabetic retinopathy, anemia, hypertension, and hyperglycemia (FPG>200 mg/dl) as predictors of incident diabetic neuropathy.

Major comments:

1. The manuscript should be written in a more fluent fashion and be more concise and comprehensive. 

Authors’ response: we tried a lot to make it concise.

There are too unnecessary repetitions, many grammatical errors, irrelevant decimals (for example instead of 179,45 months, an expression of 179 months may be easier and better for the reader). Overall, it needs an extensive editing

Authors’ response: This comment contributes a lot for the betterment of our revised manuscript. We have seen the whole document and we got a lot of redundant statements and we totally avoid irrelevant repetitions.

2. It seems that the definition of diabetic retinopathy is based on subjective, but not objective criteria (such as EMG). 

Authors’ response: we really appreciated this issue. But the diagnosis for diabetic neuropathy was done by objective criteria Nerve conduction velocity (NCV), electromyography (EMG) tests and other investigation modality by senior physians. Hence the dx for DN was not subjective.

3. What kind of anemia? Iron deficiency? Vitamin B 12 deficiency? Folate

deficiency? Chronic disease anemia? Thalassemia trait? Sickle cell trait?

Authors’ response: all types of anemia was considered as having anemia 

4. Were vitamin B12 levels measured? 

Authors’ response: really very important variable for diabetic neuropathy and at the beginning of our study, we were considered as a variable but there was no data regarding this variable.

5. What kind of oral treatments were used? Metformin? Sulfonylureas?

Pioglitazone? SGLT2 inh? DPPIV inh.?

Authors’ response: Gliben clamide was use.

6. Was not Injectable GLP-1 analogues available during the study period?

Authors’ response: was not available 

7. What kind of insulins were used? Basal, basal-bolus, mixed?

Authors’ response: all were used based on the circumstance.

8. Did the authors monitor glycemic status of the patients? For example, glycemic fluctuations are one of the major contributors to the diabetic neuropathy development. Was there hypoglycemia episodes during the study, did the authors consider this parameter when performing the analysis? 

Authors’ response: We appreciate the intension but our objective was not on measuring the glycemic control rather time to diabetic neuropathy. Despite it is one factor for the development of diabetic neuropathy, we were un able to measure the long term glycemic control due to the nature of our data source i.e. secondary data and the objective of this study was not measuring it. We strongly recommend to consider this variable for the next researchers using prospective cohort study design.

---

## [Editor Report · Decision Letter 1]

4 Apr 2023

Time to Diabetic Neuropathy and Its Predictors among Adult Type 2 Diabetes Mellitus Patients in Amhara Regional State Comprehensive Specialized Hospitals, Northwest Ethiopia, 2022: A Retrospective Follow up Study

PONE-D-22-25165R1

Dear Dr. %Bekalu Endalew Alamneh%,

We’re pleased to inform you that your manuscript has been judged scientifically suitable for publication and will be formally accepted for publication once it meets all outstanding technical requirements.

Kind regards,

James Nyirenda

Academic Editor

PLOS ONE
---

## [Editor Report · Acceptance letter]

20 Apr 2023

PONE-D-22-25165R1 

Time to Diabetic Neuropathy and Its Predictors among Adult Type 2 Diabetes Mellitus Patients in Amhara Regional State Comprehensive Specialized Hospitals, Northwest Ethiopia, 2022: A Retrospective Follow up Study 

Dear Dr. Alamneh:

I'm pleased to inform you that your manuscript has been deemed suitable for publication in PLOS ONE. Congratulations! Your manuscript is now with our production department. 

Kind regards, 

on behalf of

Dr. James Nyirenda 

Academic Editor

PLOS ONE